# *Aspergillus* Metabolome Database for Mass Spectrometry Metabolomics

**DOI:** 10.3390/jof7050387

**Published:** 2021-05-15

**Authors:** Alberto Gil-de-la-Fuente, Maricruz Mamani-Huanca, María C. Stroe, Sergio Saugar, Alejandra Garcia-Alvarez, Axel A. Brakhage, Coral Barbas, Abraham Otero

**Affiliations:** 1Centre for Metabolomics and Bioanalysis (CEMBIO), Department of Chemistry and Biochemistry, Facultad de Farmacia, Universidad San Pablo-CEU, CEU Universities, Urbanización Montepríncipe, Boadilla del Monte, 28660 Madrid, Spain; mar.mamani.ce@ceindo.ceu.es (M.M.-H.); cbarbas@ceu.es (C.B.); aotero@ceu.es (A.O.); 2Department of Information Technology, Escuela Politécnica Superior, Universidad San Pablo-CEU, CEU Universities, Urbanización Montepríncipe, Boadilla del Monte, 28660 Madrid, Spain; Sergio.saugargarcia@ceu.es (S.S.); alejandragarciaalv@gmail.com (A.G.-A.); 3Department of Molecular and Applied Microbiology, Hans Knöll Institute (HKI), Leibniz Institute for Natural Product Research and Infection Biology, Institute of Microbiology, Friedrich Schiller University Jena, 07745 Jena, Germany; maria.stroe@hki-jena.de (M.C.S.); axel.brakhage@hki-jena.de (A.A.B.)

**Keywords:** *Aspergillus*, mass spectrometry, databases, metabolomics, annotation, identification

## Abstract

The *Aspergillus* Metabolome Database is a free online resource to perform metabolite annotation in mass spectrometry studies devoted to the genus *Aspergillus*. The database was created by retrieving and curating information on 2811 compounds present in 601 different species and subspecies of the genus *Aspergillus*. A total of 1514 scientific journals where these metabolites are mentioned were added as meta-information linked to their respective compounds in the database. A web service to query the database based on *m*/*z* (mass/charge ratio) searches was added to CEU Mass Mediator; these queries can be performed over the *Aspergillus* database only, or they can also include a user-selectable set of other general metabolomic databases. This functionality is offered via web applications and via RESTful services. Furthermore, the complete content of the database has been made available in .csv files and as a MySQL database to facilitate its integration into third-party tools. To the best of our knowledge, this is the first database and the first service specifically devoted to *Aspergillus* metabolite annotation based on *m*/*z* searches.

## 1. Introduction

The genus *Aspergillus* is a group comprising both beneficial and pathogenic conidial fungi, which includes more than 300 species present in the environment [1]. Some species produce a wide range of secondary metabolites that are of high industrial and therapeutic importance, such as antibiotics and statins [2]. *Aspergillus* species usually interact with organisms in an asymptomatic manner and do not cause illness. However, some species have been shown to be responsible for several disorders in organisms with low immunity that have been massively exposed to fungal spores, such as immunocompromised humans [3,4,5,6,7,8], or plants and plant-derived products that have been exposed to mycotoxin-producing molds, leading to food contamination and spoilage [9,10,11]. Invasive aspergillosis can be among the most serious fungal infections [12,13,14,15]. Among the many species of the genus *Aspergillus*, only some are considered pathogenic. *Aspergillus fumigatus* is a major cause of disease in humans, followed by *Aspergillus flavus, Aspergillus niger, Aspergillus terreus, Aspergillus nidulans,* and several members of *Aspergillus* section *Fumigati* [13,16,17].

Aspergillosis is a disease with a high mortality rate and it is generally caused by inhalation of *Aspergillus fumigatus* conidia. These species have a highly adaptable metabolism that allows them to withstand high stress conditions, to face the immune response of the host, to acquire nutrients, or to take competitive advantage during infection. As a result of infection, the production of some fungal metabolites may undergo alterations, reflecting the fungal response to the changes brought by the host. Although aspergillosis has been extensively studied in recent decades, there is still a need for further study of the impact of host-fungal interactions [18].

Omic sciences such as genomics [19,20,21], transcriptomics [22,23,24], proteomics [25,26,27], and more recently metabolomics [28,29,30,31], as well as multi-omics approaches [32,33] have been applied to study the genus *Aspergillus*. Metabolomics is considered to be the omic that best reflects the phenotype response [34,35,36]. It generates a high volume of information about the organism in a mostly untargeted way, and it is currently one of the fastest growing research areas in the study of *Aspergillus*.

The most widely used technique to find primary and secondary metabolites, or to perform metabolomic studies, is mass spectrometry (MS) coupled with separation techniques. MS provides the mass-to-charge ratio (*m*/*z*) of the ions of interest. However, one of the main challenges in untargeted metabolomics is metabolite identification [37,38,39]. This process typically involves searching for the *m*/*z* values in various metabolomic databases, or in a mediator tool that provides a unified interface to those databases [40]. Nowadays there is no available tool to annotate *Aspergillus* compounds using *m*/*z* data obtained by MS instrumentation. There are general databases containing metabolites present in *Aspergillus* such as *Aspergillus* Secondary Metabolite Database (A2MDB) [2], KNApSAcK [41], ChEBI [42] or the Dictionary of Natural Products; there are dedicated databases such as FungiDB [43] or AspGD [44], focused on *Aspergillus* genomics data; and databases such as *Aspergillus*&Aspergillosis (https://www.Aspergillus.org.uk, accessed on 26 April 2021) devoted to general purposes. NP Atlas [45] is an open-source database containing 24,594 natural products, of which 2029 belong to the genus *Aspergillus*. However, it only collects the first isolation reference for each compound. Untargeted metabolomic studies are usually trying to annotate compounds and interpret their biological meaning. Therefore, having several references, and having the most up to date and relevant references, is valuable. Moreover, users cannot perform direct searches based on *m/z* data obtained by means of MS, but they need to perform a manual transformation of the *m*/*z* to monoisotopic mass. Although this step is trivial, it requires time, which can be reduced by automatically retrieving the compound candidates for each potential adduct formed. Furthermore, NP Atlas only permits searching for a single mass at a time, which makes the process of searching for all the masses obtained by MS quite tedious. None of the databases mentioned in this paragraph has support for an *m*/*z* search for data obtained by MS. Moreover, there are a large number of publications describing metabolites that have been identified in *Aspergillus*, but the information about many of these metabolites has not been included in any database.

The recent untargeted metabolomic studies on *Aspergillus* have used internal libraries [30] or general metabolomic databases [29,30,31] for metabolite identification. The first step in performing metabolite identification in MS studies is, in most cases, the annotation process. It consists of the assignment of putative compounds to the *m*/*z* obtained by analytical instrumentation as a necessary initial step to inferring biological significance. Currently, this step is often performed by querying general metabolomic databases. However, the number of specific *Aspergillus* metabolites in databases such as KEGG [46], LipidMaps [47], HMDB [48], or METLIN [49] remains low, which increases the chance that an *Aspergillus* metabolite is annotated as unknown. Another option is to query general databases with a much higher number of chemical compounds, such as PubChem [50] or ChEBI [42]. However, the high number of hits that are usually obtained in these databases for each *m*/*z* makes the subsequent annotation tedious work and increases the likelihood of incorrectly annotating some compounds. The use of internal libraries can be a good solution if the quality of the internal library is high, but it limits the reproducibility of the results and hinders cooperation between research groups. Therefore, the development of a tool specifically devoted to metabolite annotation in *Aspergillus* studies would provide a great value to the field.

We have integrated information about *Aspergillus* metabolites present in several metabolomic databases, including KNApSAcK, A2MDB, and ChEBI, and from scientific publications into a single database: the *Aspergillus* Metabolome Database. It comprises 2811 unique compounds that are present in 601 different species and subspecies of genus *Aspergillus*. The database also contains meta-information about metabolites, including references to scientific publications where these metabolites are mentioned, and information about the *Aspergillus* phylogenetic tree. A web tool to query the database based on *m*/*z* searches was added to CEU Mass Mediator (CMM) [51], as well as a RESTful API to enable third-party tool integration. The next section presents details on how the compounds were curated and integrated, and how the search tool was constructed. The final result of this process, together with a comparison of the number of *Aspergillus* metabolites present in the database created in this paper and in the general metabolomic databases, is presented in the results section. Finally, the utility of the service developed is discussed, concluding that it increases the *Aspergillus* metabolome coverage compared to the main metabolomic databases used in untargeted metabolomics, such as METLIN or KEGG.

## 2. Materials and Methods

### 2.1. Collection and Integration of Aspergillus Metabolites

Different *Aspergillus* species have been studied for decades and much of this information is scattered in many different resources. An effort was made to collect and organize this information, both from the already available databases, as well as by performing a bibliographic review.

The Natural Product Atlas (NP Atlas), *Aspergillus* Secondary Metabolite Database (A2MDB), KNApSAcK (a general metabolite database containing information on the species in which these metabolites are present), and metabolites compounds tagged with the *Aspergillus* role from the general metabolomics database ChEBI were selected for integration because they contain metadata that permits identifying which metabolites from these databases are present in *Aspergillus*. This metadata allowed us to build an automatic import process for metabolites (all those present in A2MDB, and those present in NP Atlas, KNApSAcK, and ChEBI related to *Aspergillus*). To this end, custom code was built in the Java language that imported files in JSON (NP Atlas), excel (A2MDB), and HTML (KNApSAcK) format using the GSON (JSON) and the java.io libraries (excel and HTML), respectively. We plan to update the compounds from these sources every 6 months. The compounds were integrated into the CMM database after completing a unification process that is described later in this section. This unification process avoids the presence of duplicates from distinct data sources. The imported information includes the name of the metabolite, its monoisotopic accurate weight, its chemical formula, its three-dimensional structure if available, and the species/species of *Aspergillus* in which the metabolite was found. When naming the different *Aspergillus* species, we adopted the nomenclature proposed in [52].

Unfortunately, general metabolic databases such as HMDB, LipidMaps, or KEGG do not contain metadata about the *Aspergillus* organisms that can be processed automatically by computational tools. Therefore, no information could be extracted from them. Some other databases with interesting information for this project, such as Dictionary of Natural Products or AntiBase, had to be discarded due to their licensing terms.

An extensive literature review has also been carried out to select the articles with information on metabolites present in *Aspergillus* (Appendix A) [53,54]. The purpose of this literature review is twofold: on the one hand, identifying new metabolites not present in the integrated databases to be added manually to the *Aspergillus* Metabolome Data, and on the other hand, to add meta-information relative to scientific publications where metabolites (both those found in the bibliographic review and those imported automatically) are discussed.

To identify metabolites not present in the integrated databases, we faced the challenge of not having well-defined search terms that would permit carrying out a systematic literature search: we are looking for metabolites that are present in *Aspergillus*, but which we do not know if they have been found in this organism. Therefore, we do not know what to look for. Although we tried to search for generic terms such as “*Aspergillus* metabolite” or “*Aspergillus* metabolome”, these searches were not effective for our purposes. The literature search started by looking for review papers related to *Aspergillus* and metabolomics that were published in indexed journals. The metabolites described in them that were not present in any of the integrated databases were incorporated into an Excel spreadsheet, together with their literature references. These references were also further explored to find more metabolites. There are no guarantees that this procedure was comprehensive (it almost certainly was not). However, as we show in the results section, this ad hoc procedure allowed us to find several hundred metabolites that were not present in any of the databases that we have integrated.

Regarding the search for literature references, when in the database from which a metabolite was obtained there was no literature reference associated, the name of the metabolite was searched for together with the word “*Aspergillus*” to find literature references that could be added to the metabolite’s meta-information. In this process, recent papers with a high number of citations were favored, given that they are more likely to provide relevant information to a researcher who has found such a metabolite in a MS study. Both in the search for literature references and in the search for new metabolites, if information about the biological activity of the compound was found in any reference, it was saved in a free text field that was added as meta-information in the *Aspergillus* Metabolome Database. The literature references and the meta-information are useful for users who can check if the putative metabolites associated with the features obtained by MS means have been previously detected in similar organisms; moreover, users can also consult the references that associate the organism with the putative identification. The complete information about compounds, including literature references and the meta-information, is provided through a RESTful API and in the MySQL backup database.

Sometimes the same metabolite is already present in NP Atlas, A2MDB, KNApSAcK, ChEBI or scientific literature under different names. Due to the existence of several names for the same metabolite, it is also possible that during the bibliographic review a metabolite was included that was already present in some of these databases under some other name. Therefore, a unification process is necessary to eliminate these duplicate entries, and to integrate the meta-information associated with each one of them.

The IUPAC International Chemical Identifier (InChI) was used for performing the unification of compounds. The InChI, in contrast to the authority-assigned identifiers such as CAS, EC Numbers, or CID from PubChem, is derived from the structural formula of the molecule. Therefore, anyone can produce the InChI for a given structure. This identifier is unique: the same InChI always corresponds to the same substance, making it ideal to achieve compound unification through different databases. The InChI can be generated using Mol files (*.mol) containing information about the three-dimensional structure of the molecule. One of the disadvantages of the InChI is that it is variable in length, which can cause complications when storing it in relational databases. The InChI Trust version 1.04 provides a Hash algorithm to generate an InChI Key, which is also unique for each substance but whose length is always 27 characters, making the identifier easier to handle.

NP Atlas, KNApSAcK, and ChEBI compounds contain their InChI. A2MDB compounds do not, neither do most compounds from the bibliographic review. When neither the InChI nor the 3D structure were available, but the name was available, we searched for the name in the CMM database, which contains 186,638 compounds. If the name appeared, and it was manually verified that it was the same compound, the InChI available in the CMM database was used for the compound. If the compound name was not available, or its name did not appear in the CMM database, its structure was drawn using ChemDraw. This tool permits exporting the structure of the molecule in different formats, including InChI. The process of InChI unification prevents the duplication of compounds in the database, provides coherence to all the metadata associated with compounds, and permits the representation of all the isomers as different metabolites (see Appendix A, a MySQL model that can be open using MySQL Workbench to navigate through all the entities present in CMM).

The classification of distinct *Aspergillus* species in a hierarchy that accurately represents the genus *Aspergillus* has the potential to permit refining *m*/*z* searches to subspecies and provide clues for the biological interpretation of the results. The relation between species and the relation between the compounds, species, and references to scientific articles permits representing useful metadata to the database users (see Appendix A). Some of the compounds are only related to the general genus *Aspergillus* and they are not classified into any species, since no studies reporting their identification in specific species have been found. The information about the biological activity of the compounds, when available, has also been included as meta-information in the database. The biological activity is a free text description based on the literature review performed.

### 2.2. Creation of the Search Tool

CMM existing infrastructure was reused for creating the search service for *Aspergillus* compounds. CMM currently runs on an Apache TomEE 7.1.4 application server and it uses the relational database MySQL Server 8.0.21. CMM entity compound was extended to model the new information regarding *Aspergillus* compounds. Three new entities were created: *Aspergillus* compounds, organisms, and references in order to represent and structure the information from each entity respectively (see Figure 1 and Appendix A). Among other advantages, this information permits categorizing the compounds in different species according to the taxonomy published by Samson et al. [48]. Freemarker templates were used to generate the web browser views to display the entities.

A new web tool was added to CMM to permit querying the databases through the *m*/*z* obtained by analytical instrumentation. Optionally, the user may specify the tolerance allowed for the precursor ion and the product ions in Da or ppm (the default value is 10 ppm). The service has all the features already implemented in CMM, including features such as the automatic identification of adducts, or the providing of scores of the likelihood of an annotation based on an expert system [51]. The search can be executed only over the *Aspergillus* Metabolome Database, or it can also include a user-selectable set of general metabolomic databases (including HMDB, LipidMaps, METLIN, KEGG, FAHFA Lipids, and MINE). The search functionality is also available through a RESTful API to facilitate its integration with third-party tools.

In order to test the web service, 26 *Aspergillus* metabolites that had previously been identified with reference standards were used (see Appendix A). The compounds were identified in *Aspergillus* samples by means of targeted analysis. To collect data about these 26 metabolites, whole culture extracts of *Aspergillus fumigatus* or *Aspergillus nidulans* were analyzed by LC-MS and compared with authentic standards. The culture broth containing fungal mycelium was homogenized using an ULTRA-TURRAX (IKA-Werke, Staufen, Germany). Homogenized cultures were extracted twice with a total of 100 mL ethyl acetate, dried with sodium sulfate, and concentrated under reduced pressure. For LC-MS analysis, the dried extracts were dissolved in 1 mL of methanol, while the authentic standards were prepared at a concentration of 1 mg/mL. The samples were loaded onto an ultrahigh-performance liquid chromatography (LC)-MS system consisting of an UltiMate 3000 binary rapid separation liquid chromatograph with photodiode array detector (Thermo Fisher Scientific, Dreieich, Germany) and an LTQ XL linear ion trap mass spectrometer (Thermo Fisher Scientific, Dreieich, Germany) equipped with an electrospray ion source. The extracts or the authentic standards (injection volume of 10 μL) were analyzed on a 150 mm by 4.6 mm Accucore reversed-phase (RP)-MS column with a particle size of 2.6 μm (Thermo Fisher Scientific, Dreieich, Germany) at a flow rate of 1 mL/min, with the following gradient over 21 min: initial 0.1% (*v*/*v*) HCOOH-MeCN/0.1% (*v*/*v*) HCOOH-H_2_O 0/100, which was increased to 80/20 in 15 min and then to 100/0 in 2 min, held at 100/0 for 2 min, and reversed to 0/100 in 2 min. These 26 metabolites were identified by comparison with reference standards. To test the service, we searched their *m*/*z*s in both CMM and METLIN, the general metabolomic database providing an *m*/*z* search which is the database containing the largest number of *Aspergillus* compounds. For comparison, we also searched them one by one in NP Atlas using the monoisotopic masses.

## 3. Results

The *Aspergillus* Metabolome Database comprises 2029 metabolites from NP Atlas, 614 metabolites from KNApSAcK, 700 compounds from A2MDB that were manually curated, and 491 compounds from the bibliography review, which included 337 papers. After the unification of compounds based on the InChI, the database contains 2811 unique compounds. The complete database is publicly available for download as a MySQL backup, and as three Excel files (https://github.com/albertogilf/ceuMassMediator/tree/master/CMMAspergillusDB, accessed on 26 April 2021).

The full content of the *Aspergillus* Metabolome Database, including metadata, was added to the CMM database. This resulted in the addition of 2124 new compounds to the CMM database and the inclusion of metadata for other 203 compounds already present in the database. Three new resources were created in the CMM database to provide support for these compounds. The first resource named “compound” contains information about the compounds (Figure 2a). It includes structural information such as formula, mass, charge type, charge number, InChI, InChI Key, SMILES, and IUPAC Classification; links to other databases such as CAS registry, ChEBI, HMDB, KEGG, KNApSAcK, METLIN, and PubChem; the classification from ClassyFire tool [55]; and the organisms (*Aspergillus* species and subspecies), ontology terms, and literature references that the compound is linked to. The second resource named “reference” represents the literature references and contains a list of organisms and compounds that are mentioned in the publication (Figure 2b). The last resource represents a specific organism and shows the list of compounds known to be present in the specie and the literature references where these compounds or the species are described (Figure 2c). Information about 2811 compounds associated with 220 *Aspergillus* species and 381 subspecies, and links to 1514 literature references where these are mentioned, can be consulted through the web resources that were added to CMM (Figure 2a).

This information was included in the MS searches available at CMM, both in the web search tool and in the RESTful API services. A JSON representation of the new entities is available using CMM’s API services. A new CMM Card, that is, a web representation of the information of a metabolite, was created to adequately display the new information present in *Aspergillus* compounds (Figure 2a). The service to annotate *Aspergillus* metabolites is freely available through the metabolite annotation tool CMM (http://ceumass.eps.uspceu.es/, accessed on 26 April 2021). The source code of the *Aspergillus* service is available in the CMM Github code repository: https://github.com/albertogilf/ceuMassMediator/tree/master/ceu-mass-mediator-v4.0, accessed on 26 April 2021. The web application for *Aspergillus* metabolite annotation is available at http://ceumass.eps.uspceu.es/mediator/Aspergillus_metabolome_search.xhtml, accessed on 26 April 2021.

The number of metabolites present in the *Aspergillus* Metabolome Database that are also present in the general metabolomic databases providing batch *m*/*z* searches (KEGG, LipidMaps, HMDB, and METLIN) are shown in Table 1, while the coverage between natural products-databases not providing *m*/*z* searches is shown in Table 2. In untargeted metabolomics, the annotation of compounds is key and a batch *m*/*z* search is usually the first step to annotate, followed by searches in databases containing information about *Aspergillus* compounds. For untargeted metabolomics researching about *Aspergillus* the possibility of connecting the two types of databases seems a valuable service. Note that these general databases do not have computer-readable metadata about which metabolites are present in some *Aspergillus* species. Hence, to construct these tables, we searched for the metabolites present in the databases that also were present in the *Aspergillus* Metabolome Database using their InChI, or the name of the compound when the InChI was not available in the database. A Venn diagram showing the overlap between *Aspergillus* metabolites present in the *Aspergillus* Metabolome Database, KEGG, LipidMaps, HMDB, and METLIN is shown in Figure 3. The monoisotopic masses distribution of the 491 compounds that were not present in other databases is show in Figure 4, and the top 30 classes that they belong to according to ClassyFire are shown in Table 3.

Regarding the 26 *Aspergillus* compounds previously identified by comparison with authentic standards, all of them were present in the *Aspergillus* Metabolome Database, 12 were present in NP Atlas, and 8 were present in METLIN. The results can be seen in the Appendix A.

## 4. Discussion and Conclusions

The dispersion in the distinct data sources is a general issue in metabolomics. In the case of *Aspergillus*, this dispersion was compounded with a large number of references generating knowledge about *Aspergillus* which were not collected in metabolomic databases. Both these situations hinder the annotation process and slow down research on *Aspergillus*.

The work presented here tackles this problem by creating the *Aspergillus* Metabolome Database. This database was created by compiling metabolites present in *Aspergillus* from public databases whose license allowed their integration into a third-party database (NP Atlas, KNApSAcK, A2MDB, and ChEBI), as well as through a literature review that included 337 articles. These compounds were unified by means of their InChI to avoid duplications. After the unification process, the database contains 2811 unique *Aspergillus* metabolites and 1514 papers with information on those metabolites. In addition to the name of the compound, its formula, and its *m*/*z*, the database contains meta-information, such as taxonomic information on the different *Aspergillus* species, and references to publications related to the compound.

A comparative analysis of the number of *Aspergillus* metabolites contained by the general metabolomic databases was performed. The general database that contained the highest number of *Aspergillus* compounds present in the *Aspergillus* Metabolome Database was METLIN, which contained 214 compounds. This means that approximately 92% of the compounds available in *Aspergillus* Metabolome Database were not present in METLIN. Even if we consider all general metabolomic databases together, 2535/2811 (90%) of the compounds present in the *Aspergillus* Metabolome Database were not present in any of them (see Table 1 and Figure 3). This suggests that if the general metabolomic databases are used to annotate *Aspergillus* compounds, it is highly likely that an *m*/*z* corresponding to a known compound returns no hits, and results in an annotation of the compound as unknown, even when searching for the *m*/*z* in all of them.

Due to the lack of metadata regarding which compounds are present in some *Aspergillus* species in the general metabolomic databases, it is not possible to know if there are compounds identified in *Aspergillus* species present in them that have not been included in the *Aspergillus* Metabolome Database. This also means that, if such compounds existed, it is not possible to include them in the *Aspergillus* Metabolome Database. However, the fact that most of the compounds present in the *Aspergillus* Metabolome Database are not present in these databases (see Table 1 and Figure 3) leads us to believe that if such compounds exist, they should be very few.

A web tool was built and integrated into CMM that permits searching for *m*/*z* over the *Aspergillus* Metabolome Database, specifying the tolerance of the mass, the possible adducts formed, the chemical alphabet, the modifiers used, and the inclusion or exclusion of general databases. This functionality is also available through a RESTful API, which can permit its integration of *m*/*z* searches of *Aspergillus* compounds in third-party tools. The source code for the web search tool and for the RESTful API has been released under an open license. In addition, all the current 2811 metabolites of the *Aspergillus* Metabolome Database have been published as Excel files and as MySQL backup files.

However, new *Aspergillus* metabolites are continually being discovered. We plan to continue updating the *Aspergillus* Metabolome Database with this information in the future, and we strongly encourage the scientific community to contribute information from internal databases on *Aspergillus* metabolites to this project. We believe that having a unified open repository for all *Aspergillus* metabolites, as well as tools that permit querying it, is a benefit for the entire scientific community involved in studying *Aspergillus* species.

## Figures and Tables

**Figure 1 jof-07-00387-f001:**
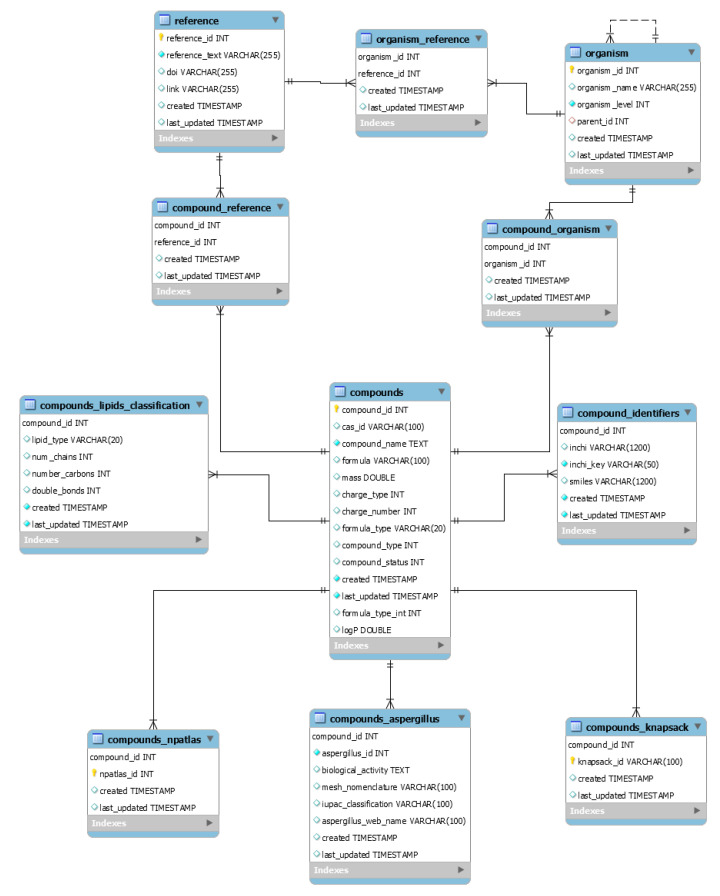
Abbreviated Entity-Relation model for the CMM *Aspergillus* Database. Full model is available in Appendix A.

**Figure 2 jof-07-00387-f002:**
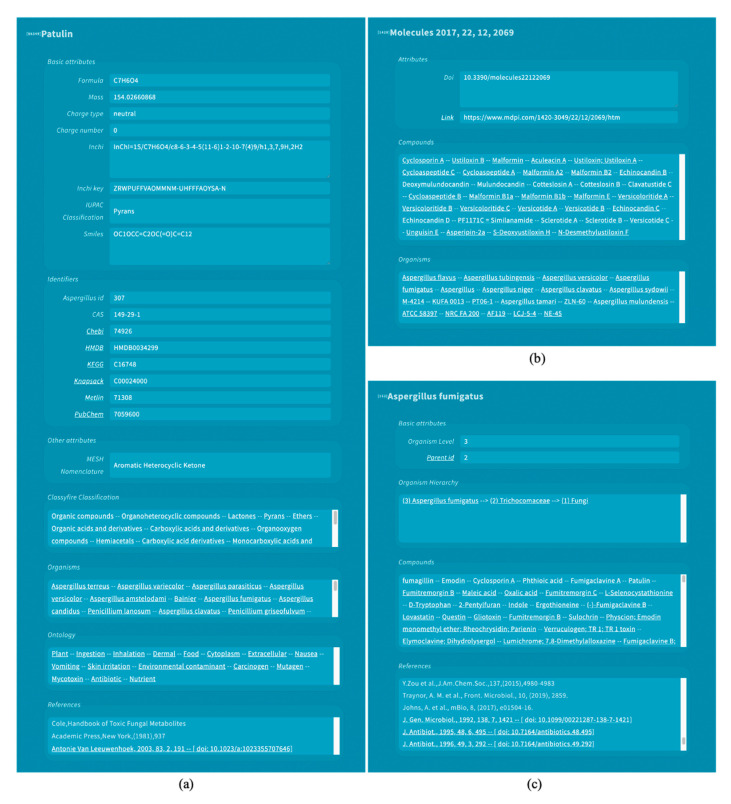
Web interface of CMM showing the information about (**a**) patulin compound; (**b**) literature reference with doi: 10.3390/molecules22122069; (**c**) *Aspergillus fumigatus* species.

**Figure 3 jof-07-00387-f003:**
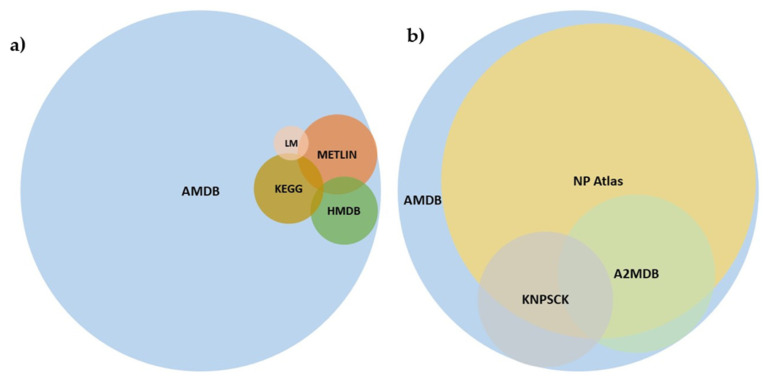
(**a**) Venn diagram of *Aspergillus* compounds present in general metabolomic databases. A, AMBD: *Aspergillus* Metabolome Database; KEGG: KEGG; LM: LipidMaps, HMDB: Human Metabolome Database; METLIN: METLIN; (**b**) Venn diagram of *Aspergillus* compounds present in databases devoted to Natural Products or *Aspergillus*. AMDB: *Aspergillus* Metabolome Database; NP Atlas: Natural Products Atlas; KNPSCK: KNApSAcK; A2MDB: *Aspergillus* Secondary Metabolites Database.

**Figure 4 jof-07-00387-f004:**
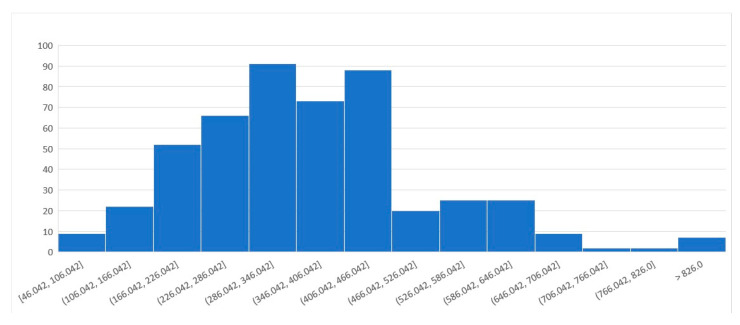
Monoisotopic masses distribution of the 491 compounds retrieved from the literature reference that were not present in NP Atlas or KNApSAcK. Y axis: number of compounds; X axis: *m*/*z* range.

**Table 1 jof-07-00387-t001:** Number of *Aspergillus* compounds in *Aspergillus*DB, CMM, KEGG, LipidMaps, HMDB, and METLIN.

AspergillusDB	KEGG	LipidMaps	HMDB	METLIN
2811	164	57	160	214

**Table 2 jof-07-00387-t002:** Number of *Aspergillus* compounds in *Aspergillus*DB, NP Atlas, KNApSAcK, and A2MDB.

AspergillusDB	NP Atlas	KNApSAcK	A2MDB
2811	2029	621	807

**Table 3 jof-07-00387-t003:** Top 30 main classes from Classyfire.

Main Class	Classyfire Node ID	Number of Compounds
Organooxygen compounds	CHEMONTID:0000323	430
Organic oxides	CHEMONTID:0003940	378
Oxacyclic compounds	CHEMONTID:0004140	245
Carboxylic acids and derivatives	CHEMONTID:0000265	239
Phenols	CHEMONTID:0000134	172
Organonitrogen compounds	CHEMONTID:0000278	152
Vinylogous acids	CHEMONTID:0003889	142
Benzene and substituted derivatives	CHEMONTID:0002279	136
Azacyclic compounds	CHEMONTID:0004139	136
Heteroaromatic compounds	CHEMONTID:0004144	131
Lactones	CHEMONTID:0000050	114
Pyrans	CHEMONTID:0000086	91
Benzopyrans	CHEMONTID:0000123	84
Lactams	CHEMONTID:0000160	78
Phenol ethers	CHEMONTID:0002341	73
Dihydrofurans	CHEMONTID:0001983	54
Fatty Acyls	CHEMONTID:0003909	49
Indoles and derivatives	CHEMONTID:0000211	48
Prenol lipids	CHEMONTID:0000259	43
Naphthalenes	CHEMONTID:0000023	42
Pyrroles	CHEMONTID:0000090	37
Vinylogous esters	CHEMONTID:0003891	35
Propargyl-type 1,3-dipolar organic compounds	CHEMONTID:0003633	33
Naphthopyrans	CHEMONTID:0001640	32
Pyrrolidines	CHEMONTID:0000218	29
Carboximidic acids and derivatives	CHEMONTID:0002285	29
Tetrahydrofurans	CHEMONTID:0002648	27
Coumarans	CHEMONTID:0004189	25
Epoxides	CHEMONTID:0000159	25
Oxanes	CHEMONTID:0002012	25

## Data Availability

Software is a public open-source repository stored at https://github.com/albertogilf/ceuMassMediator/tree/master/ceu-mass-mediator-v4.0. (Accessed on 14 May 2021), https://github.com/albertogilf/ceuMassMediator/tree/master/CMMAspergillusDB (Accessed on 14 May 2021) contains the Appendix A.

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
