# Peer review of "Aspergillus Metabolome Database for Mass Spectrometry Metabolomics"

_jof, 2021, doi:10.3390/jof7050387_

Round 1

Reviewer 1 Report

The manuscript by Gil-de-la-Fuente et al. did substantially improve with the inclusion of the NPAtlas dataset.  Furthermore,  the goal of the study is also a bit more sharply defined now. Also, this reviewer would like to highlight appreciation for sharing the code, database contents, and supplementary files as open science. This reviewer thinks that the manuscript could be made fit for publication, provided that the following points will be addressed by the authors:  

Main points:  

Literature search - the authors should provide more details on how the literature search was done. I.e., were titles and abstracts screened or the entire papers? What kind of search terms were used? And what was recorded from each paper? Do the authors aim to be comprehensive in terms of finding all related papers to a specific metabolite? Were the authors also looking for the first appearance of a metabolite in literature? (and was that recorded)? And are they also planning to repeat this literature search every so often? Given the central place of the literature research in the presented work (that is where most of the novelty and impact is coming from), this reviewer could find far too little information on the above-mentioned aspects....and the method is currently not reproducible, nor it is clear how comprehensive the search was....   

Aspergillus metabolite overview - the authors collected nearly 600 metabolites from literature that were not yet present in NPAtlas (and could be added?! - especially if the first isolation source was found); however, they provide little to no information about their structures and chemical classes. In fact, no structure is visible in the manuscript! For researchers (and also reviewers!), to assess whether it makes sense to use the new AspergillusDB resource and its potential impact, overviews of which chemical classes (i.e., by using ClassyFire and/or NPClassifier),  and the m/z distribution, and other relevant information is essential. This would also help to sell your work.   Aspergillus species overview - could the authors provide a Figure that shows the species diversity currently covered in AspergillusDB? That could replace the Venn diagrams (see remark below).  

Choice of databases - the authors mention and use various databases that include several types: spectral (i.e., MetLin), compound (i.e., NPAtlas, A2MDB), and pathway-central (i.e., KEGG). It is i) unclear how that selection came about and the authors should motivate their choice, and ii) the authors should provide some more detail on the different types.  

Minor points:    

NPAtlas (page 2) - the authors describe that NPAtlas "only collects a single literature reference for each compound" ; whilst that is true, this is intended as curators have looked for the first isolation reference  of that compound - the authors should rephrase this in their manuscript.  

(page 2) "Manual transformation of the m/z to monoisotopic mass" - Is that really a bottleneck?! This reviewers knows that there are many workflows (rdkit, cdk, etc.) now that take in elemental formulas and produce the expected monoisotopic masses in an automated manner. This is not a strong argument to make your work stand apart.... The batch search and the additional literature references and metabolites you have found are.  

Page 3 Aspergillus Secondary Metabolite Database (A2MDB), KNApSAcK  - the A2MDB database needs a reference! This reviewer could not find it. How many metabolites are in there?  

Figure 2 - the added value of the Venn diagrams on top of the Tables is little. Also, the right hand-side Venn diagram does not seem to be in the correct ratio: the difference between NPAtlas and AspergillusDB is 600 out of 2800, but it looks much larger in the diagram....

Author Response

The manuscript by Gil-de-la-Fuente et al. did substantially improve with the inclusion of the NPAtlas dataset.  Furthermore,  the goal of the study is also a bit more sharply defined now. Also, this reviewer would like to highlight appreciation for sharing the code, database contents, and supplementary files as open science. This reviewer thinks that the manuscript could be made fit for publication, provided that the following points will be addressed by the authors:  

Thank you very much for your comments. All your suggestions have helped to improve the quality of the manuscript. We have addressed the suggestions and we acknowledge your time reviewing the manuscript. Your comments pointed out certain weaknesses that we hope that are now clearer for the readers.

We do not intend to create a comprehensive database for Aspergillus metabolites. There already exists an established consortium, NP Atlas, that the community can use to submit new compounds. We intend to provide a service that connects MS-untargeted metabolomics (more specifically, the metabolite annotation process) with databases like KNApSAcK, NP Atlas and the information that we have collected from literature references that contains valuable information about Aspergillus. Several of the issues raised by the reviewer are due to the original version of the paper not making this point clear. Some of our statements were confusing because they were made considering that AspergillusDB is a tool for metabolite annotation in MS-untargeted metabolomics studies, but this point had not been made clear in the paper.

We will update information from the data sources every 6 months as we do with the general metabolomic databases, including new information that we find from literature references. We also plan to submit all the compounds found in the literature review to the NP Atlas consortium. But first we want to organize the information to facilitate the curation by the consortium.

Main points:  

Literature search - the authors should provide more details on how the literature search was done. I.e., were titles and abstracts screened or the entire papers? What kind of search terms were used? And what was recorded from each paper? Do the authors aim to be comprehensive in terms of finding all related papers to a specific metabolite? Were the authors also looking for the first appearance of a metabolite in literature? (and was that recorded)? And are they also planning to repeat this literature search every so often? Given the central place of the literature research in the presented work (that is where most of the novelty and impact is coming from), this reviewer could find far too little information on the above-mentioned aspects....and the method is currently not reproducible, nor it is clear how comprehensive the search was....   

Thank you for this comment. A systematic literature search is commonly done to perform a research about the state of the art where it is possible to define a set of search terms that can be used to systematically explore the literature references. However, the characteristics of our search were not very well suited for a systematic review. In our case, we are looking for metabolites that are present in Aspergillus, but we do not know what metabolites they are. In a way, we do not know what we are looking for. Although we tried to search for terms such as "Aspergillus metabolite" or "Aspergillus metabolome”, they were not effective for our purpose.

The bibliographic search process started by looking for review articles related to Aspergillus and metabolomics that were published in indexed journals. The metabolites present directly in these review articles not present in the databases that we have integrated were incorporated into an Excel sheet, together with their bibliographic references. These bibliographic references were also further explored to find more metabolites. In general, the biographical search process was guided by starting from these review articles, and “pulling the thread” to discover new metabolites. We cannot at all say that the process was exhaustive and that we have found all the metabolites. We can only affirm that this has allowed us to find 491 metabolites that were not present in any of the databases that we have integrated.

Regarding the search for literature references for metabolites, when in the database from which a metabolite was obtained there was no literature reference regarding it, the name of the metabolite was searched together with the word "Aspergillus" to search for literature references that could be added to the metabolite. In this process, papers with a high number of citations were favored, rather than the first paper citing the metabolite. Therefore, we cannot affirm that the references we have included are the firsts, nor that they are an exhaustive list of references regarding that metabolite. The purpose of these references is not attribution to the discovery of the metabolite, but rather to provide some information relevant to a researcher who has found such a metabolite in a MS study.

We have clarified these points it in the manuscript (lines 154-182):

“To identify metabolites not present in the integrated databases, we faced the challenge of not having well-defined search terms that would permit carrying out a systematic literature search: we are looking for metabolites that are present in Aspergillus, but which we do not know that they have been found in this organism. Therefore, we do not know what to look for.  Although we tried to search for generic terms such as "Aspergillus metabolite" or "Aspergillus metabolome”, they were not effective for our purposes. The literature search started by looking for review papers related to Aspergillus and metabolomics that were published in indexed journals. The metabolites decribed in them that were not present in any of the integrated databases were incorporated into an Excel spreadsheet, together with their literature references. These references were also further explored to find more metabolites. There are no guarantees that this procedure was comprehensive (it almost certainly was not). However, as we shall show in the results section, this ad hoc procedure allowed us to find several hundred metabolites that were not present in any of the databases that we have integrated.

Regarding the search for literature references, when in the database from which a metabolite was obtained there was no literature reference associated, the name of the metabolite was searched for, together with the word "Aspergillus" to find literature references that could be added to the metabolite's meta-information. In this process, recent papers with a high number of citations were favored, given that they are more likely to provide relevant information to a researcher who has found such a metabolite in a MS study. Both in the search for literature references, and in the search for new metabolites, if information about the biological activity of the compound was found in any reference, it was saved in a free text field that was added as meta-information in the Aspergillus Metabolome Database. The literature references and the meta-information are useful for users who can check if the putative metabolites associated with the features obtained by MS means have been previously detected in similar organisms; moreover, users can also consult the references that associate the organism with the putative identification. The complete information about compounds, including literature references and the meta-information, is provided through a RESTful API and in the MySQL backup database.”

Aspergillus metabolite overview - the authors collected nearly 600 metabolites from literature that were not yet present in NPAtlas (and could be added?! - especially if the first isolation source was found); however, they provide little to no information about their structures and chemical classes. In fact, no structure is visible in the manuscript! For researchers (and also reviewers!), to assess whether it makes sense to use the new AspergillusDB resource and its potential impact, overviews of which chemical classes (i.e., by using ClassyFire and/or NPClassifier),  and the m/z distribution, and other relevant information is essential. This would also help to sell your work.   Aspergillus species overview - could the authors provide a Figure that shows the species diversity currently covered in AspergillusDB? That could replace the Venn diagrams (see remark below).  

Thank you very much for this comment. As we have previously discussed, our goal is to provide a metabolite annotation tool, and the compounds will be submitted to the NP Atlas consortium as soon as we organize the information to facilitate the curation by their side. We also believe that it may be useful to publish this manuscript first so that we can send them a peer-reviewed paper that explains the process by which the metabolites were obtained. From the point of view of metabolite annotation, finding the first source of isolation is not critical. The important thing is to know that this metabolite is present in Aspergillus, and to provide bibliographic references that help to understand the biological role of this metabolite. From this point of view, a more recent paper with more updated and complete information about the metabolite is more relevant than the first paper where this metabolite is isolated.

Although the information about structures and chemical classes is very useful, showing the structure of 491 metabolites not present in KNApSAcK or NP Atlas provides little information to researchers performing untargeted metabolomics and the length of the manuscript would be too long. The information can be found in Supplementary Information in SMILES, InChI and InChI Key format (CSV files or MySQL dumps). The main goal of this manuscript is to provide a service for untargeted metabolomics researching about Aspergillus and the researchers can also check the structures later on in the web service.

Following your recommendations, we have included a histogram with the distribution of the monoisotopic masses of the compounds identified in the literature search (Figure 4). We have also added a table (Table 3) with the top 30 sub classes that the new compounds belong according to classyFire taxonomy.

Choice of databases - the authors mention and use various databases that include several types: spectral (i.e., MetLin), compound (i.e., NPAtlas, A2MDB), and pathway-central (i.e., KEGG). It is i) unclear how that selection came about and the authors should motivate their choice, and ii) the authors should provide some more detail on the different types.  

This comment is again very useful. The information in the manuscript was not clear and we have rephrased it in page 8. Although the databases contain different information, we have classified them based on 1) databases containing information about all type of compounds but providing a batch m/z search and 2) natural product databases containing information about Aspergillus compounds but not providing a batch m/z search.

We have rephrased it in lines 313-329 to:

“The number of metabolites present in the Aspergillus Metabolome Database that are also present in the general metabolomic databases providing batch m/z searches (KEGG, LipidMaps, HMDB and METLIN) are shown in Table 1, while the coverage between natural products-databases not providing m/z searches are shown in Table 2. In untargeted metabolomics, the annotation of compounds is key and a batch m/z search is usually the first step to annotate, followed by searches in databases containing information about Aspergillus compounds. For untargeted metabolomics researching about Aspergillus the possibility of connecting the two types of databases seems a valuable service. Note that these general databases do not have computer-readable metadata about which metabolites are present in some Aspergillus species. Hence, to construct these tables, we searched for the metabolites present in the databases that also were present in the Aspergillus Metabolome Database using their InChI, or the name of the compound when the InChI was not available in the database. A Venn diagram showing the overlap between Aspergillus metabolites present in the Aspergillus Metabolome Database, KEGG, LipidMaps, HMDB, and METLIN is shown in Figure 3.”

Minor points:    

NPAtlas (page 2) - the authors describe that NPAtlas "only collects a single literature reference for each compound" ; whilst that is true, this is intended as curators have looked for the first isolation reference  of that compound - the authors should rephrase this in their manuscript.  

Thank you for pointing this out. As we already have mentioned, in untargeted metabolomic studies, having the most up-to-date and complete metabolite information to understand the biological role of a metabolite is more important than having the first reference. From this point of view, a more recent paper with more updated and complete information about the metabolite is more relevant than the first paper where this metabolite is isolated. Furthermore, the researchers often try to find connections between the compounds and one or more organisms, therefore having several references is valuable for them. We have explained that in the lines 71-75:

“NP Atlas [45] is an open-source database containing 24,594 natural products, of which 2,029 belong to the genus Aspergillus. However, it only collects the first isolation reference for each compound. Untargeted metabolomic studies are usually trying to annotate compounds and interpret their biological meaning. Therefore having several references, and having the most up to date and relevant references, is valuable.“

(page 2) "Manual transformation of the m/z to monoisotopic mass" - Is that really a bottleneck?! This reviewers knows that there are many workflows (rdkit, cdk, etc.) now that take in elemental formulas and produce the expected monoisotopic masses in an automated manner. This is not a strong argument to make your work stand apart.... The batch search and the additional literature references and metabolites you have found are.  

Thank you again for this comment. The calculation of expected monoisotopic masses is trivial, but in most of the untargeted studies is not possible to get a single formula for the different m/zs. The possibility of introducing a single m/z and retrieve all the putative candidates for the possible adducts formed is very useful, especially for novel researchers. We have included in lines 78-80 the sentence:

"Although this step is trivial, it requires time, which can be reduced by automatically retrieving the compound candidates for each potential adduct formed.”

Page 3 Aspergillus Secondary Metabolite Database (A2MDB), KNApSAcK  - the A2MDB database needs a reference! This reviewer could not find it. How many metabolites are in there?  

Thank you very much for this comment. We cited the A2MDB in the introduction (reference 2) (line 67).

Figure 2 - the added value of the Venn diagrams on top of the Tables is little. Also, the right hand-side Venn diagram does not seem to be in the correct ratio: the difference between NPAtlas and AspergillusDB is 600 out of 2800, but it looks much larger in the diagram....

We agree with the reviewer that the information of the Venn diagrams on top of the Tables is little. However, there are researchers that want to know in a diagonal read what is the coverage of 1) Aspergillus compounds in general untargeted metabolomic databases (figure 3a) and 2) databases devoted to Natural Products or Aspergillus (Figure 3c). We have redone the Venn Diagrams in such a way that the size and overlap of the circles correctly reflects the overlap between the databases.

Reviewer 2 Report

The manuscript could be accept in  present form

Author Response

Dear reviewer,

thank  you very much for your previous comments. We acknowledge your time reviewing the manuscript and we inform you that we have made some changes in the manuscript to improve its quality according to other reviewers' comments. We have detailed the bibliography review and we have included information about the monoisotopic distribution as well as the ClassyFire main classes that new compounds not present in other databases belong to. All these changes have been updated in a word document with track changes.

Thank you again,

The authors

Round 2

Reviewer 1 Report

The manuscript by Gil-de-la-Fuente et al. did again substantially improve by better descriptions of the methodology, and a better representation of the newly found Aspergillus metabolites. Furthermore,  the goal of the study is again more sharply defined. Based on this reviewers' expertise, the manuscript is technically sound. Having said that, the English language could still be improved at several places to make everything (even) more understandable (and convincing).

This reviewer would like to stress that the authors are highly encouraged to follow their plans in adding the newly found Aspergillus metabolites to NP Atlas and looks forward to see them appearing there. This reviewer agrees that having this manuscript (that now includes the methodology followed to collect the Aspergillus metabolites) published will help in the process of merging these two nice resources.

This manuscript is a resubmission of an earlier submission. The following is a list of the peer review reports and author responses from that submission.

Round 1

Reviewer 1 Report

In the manuscript ‘Aspergillus Metabolome Database for Mass Spectrometry Metabolomics’ the authors have provided new database, named Aspergillus Metabolome Database integrating information about Aspergillus metabolites present in metabolomic databases, including KNApSAcK, A2MDB and ChEBI, and scientific publications. Genus Aspergillus was deeply investigated in order to clarify the impact of host-fungal interactions.

Moreover, the database also contains meta-information about metabolites, including references to scientific publications where these metabolites are mentioned, and information about the Aspergillus phylogenetic tree.

  1. The databases construction based on correct and clean workflow
  2. Aspergillus metabolites collection was accurate

Please provide a real example of using the databases

Author Response

In the manuscript ‘Aspergillus Metabolome Database for Mass Spectrometry Metabolomics’ the authors have provided new database, named Aspergillus Metabolome Database integrating information about Aspergillus metabolites present in metabolomic databases, including KNApSAcK, A2MDB and ChEBI, and scientific publications. Genus Aspergillus was deeply investigated in order to clarify the impact of host-fungal interactions.

Moreover, the database also contains meta-information about metabolites, including references to scientific publications where these metabolites are mentioned, and information about the Aspergillus phylogenetic tree.

  1. The databases construction based on correct and clean workflow
  2. Aspergillus metabolites collection was accurate

Please provide a real example of using the databases

Dear reviewer, thank you very much for your comments. We have provided an example of using the database. We have taken a set of features from 26 reference standards previously detected in Aspergillus nidulans and Aspergillus fumigatus and we have identified them using different general metabolomic databases providing m/z searches and the Aspergillus Metabolome database.

Best regards,

The authors.

Reviewer 2 Report

The manuscript by Gil-de-la-Fuente et al. describes the introduction of a metabolomics database targeting Aspergillus fungal strains within their CMM framework. Whilst this reviewer can see the potential use cases, there are three general points that the authors will need to work on. Most importantly, the authors do not mention or refer to the NPAtlas, a curated resource for specialized metabolites from microorganisms. For example, 2034 Aspergillus metabolites are recorded there, which is twice the amount of the metabolites the authors report. How is that possible? Without a good explanation and clear idea of the contribution of this paper [this reviewer makes some suggestions regarding the contribution further below - see points on application and bioactivity], the justified publication of this manuscript will be difficult. Furthermore, an example application of the database is lacking. This reviewer made some further comments and suggestions for clarifications of various points. Finally, the English language should be looked at throughout the entire manuscript but especially the abstract and discussion and conclusion section.

General comments:

[from introduction] "Nowadays there is no available tool to annotate Aspergillus compounds using m/z data obtained by MS instrumentation."
--> This statement is incorrect! The NPAtlas (
https://www.npatlas.org/joomla/index.php --> https://pubs.acs.org/doi/10.1021/acscentsci.9b00806) provides 2034 manually curated Aspergillus metabolites with their elemental formulas and accurate masses. This is also double the amount of the database the authors provide here. Thus, this resource needs to be cited in the manuscript and the authors will need to carefully explained how their database is different from what NPAtlas offers (what is the novelty [i.e., the web-tool to search for the metabolites], and why not simply integrate the resources to prevent database fragmentation and retrieve the metabolites from NPAtlas). How many metabolites are reported by the authors and not by NPAtlas? Also, the authors could add the missing Aspergillus metabolites in their database and vice versa? Also, add NPAtlas into the Venn Diagram of Figure 3.

Application of the new database:
--> Whilst this reviewer can see the potential use cases of the database (whether or not integrated with NPAtlas), showing an example of how it is applied to a fungal extract seems lacking from the manuscript. It  would also make a lot of things implicitly mentioned in the manuscript more explicit. This would also add a nice Figure where a chromatogram with numerous annotated Aspergillus metabolites could be displayed that would not have had useful candidates in other open databases. The authors could take one dataset of their own work, or get a public dataset run on Aspergillus extracts. If MS/MS data of some or several features is available, those annotations could be verified as well and added as novel database compounds to public spectral libraries - which would constitute as a clear contribution from this manuscript. It would serve as an input for a nice graphical summary as well.

English language
--> Language needs polishing at several instances - especially grammar (use of present and paste tense). Especially so in the abstract as well as throughout the manuscript.

Introduction

"Aspergillus fumigatus is the 42 major cause of disease in humans followed by Aspergillus flavus, Aspergillus niger, Aspergillus terreus, Aspergillus nidulans and several members of Aspergillus section Fumigati [13,16,17]." 
--> How was this ranking done? And by whom? By the authors? This needs some clarification in the manuscript.

"Metabolomics is considered the omic"  --> Rephrase: Metabolomics is considered to be the omics discipline  

"However, one of the main challenges in untargeted metabolomics is metabolite identification." 
--> this statement needs some references.

"Finally, the results of this paper are discussed and conclusions are drawn."
--> instead of this "empty" statement, the authors could actually shortly summarize the main results, findings, and conclusions here....

Material and Methods

"to add metainformation relative to scientific publications where metabolites (both those found in the bibliographic review and those imported automatically) are discussed. Among the included metainformation from the literature review are also associations between metabolites and the Aspergillus species where they are present." 
--> was this metainformation also computer-readable stored? More detailed information at this stage on the actual content, how it could be used, and the purposes, is needed here.

"we searched for the name in the 164 CMM database, which contains 186,638 compounds."
--> this needs a ref to the "CMM" database. And also explain the abbreviation "CMM" here!

"(see Supplementary Information, file “SI1_fullERModel.mwb”)"
--> it is unclear what is present in this file,  or how it can be opened.

"The information about the biological activity of the compounds, when available, has also been included as meta-information in the database."
--> How was this done? Did the authors make several categories? Was a particular ontology used here? And is it computer-readable? Can you search based on bioactivity? Could you provide an overview in the manuscript of this? This could be of added value to the community by linking the activity to the structures and then to omics data obtained from those structures.

Figure 3
--> The Venn diagram implies that some databases contain "unique" Aspergillus metabolites (i.e., on the right hand side the circles are outside the main large circle), whereas in practice all metabolites are part of the author's database, right? This should be clarified as it is confusing now.
In addition, the Figure is not that "space-efficient" and the authors could consider a different representation of the same information.

Discussions and conclusions

"Therefore, we cannot include these potential Aspergillus compounds in the Aspergillus Metabolome Database (note that if this calculation was possible, the automatic incorporation of these compounds into the Aspergillus Metabolome Database would also be possible)."
--> What do the authors mean with "calculation" here? This is unclear and needs rephrasing.

Author Response

The manuscript by Gil-de-la-Fuente et al. describes the introduction of a metabolomics database targeting Aspergillus fungal strains within their CMM framework. Whilst this reviewer can see the potential use cases, there are three general points that the authors will need to work on. Most importantly, the authors do not mention or refer to the NPAtlas, a curated resource for specialized metabolites from microorganisms. For example, 2034 Aspergillus metabolites are recorded there, which is twice the amount of the metabolites the authors report. How is that possible? Without a good explanation and clear idea of the contribution of this paper [this reviewer makes some suggestions regarding the contribution further below - see points on application and bioactivity], the justified publication of this manuscript will be difficult. Furthermore, an example application of the database is lacking. This reviewer made some further comments and suggestions for clarifications of various points. Finally, the English language should be looked at throughout the entire manuscript but especially the abstract and discussion and conclusion section.

Thank you very much for your comments. Honestly, we did not know about NP Atlas. Our background is on Metabolomics and Mass Spectrometry and NP Atlas did not appear in our original literature review. This suggestion has improved considerably the purpose of the Aspergillus Metabolome Database; after integrating NP Atlas the number of Aspergillus metabolites present in our database has increased up to 2,811. NP Atlas is a very useful resource to collect data from Natural Products and our tool is a useful resource to perform metabolite annotation. The purpose of the manuscript is to provide a useful tool to perform metabolite annotation in Aspergillus studies. Hence, our service is a complement to the NPAtlas database, not an alternative.

General comments:

[from introduction] "Nowadays there is no available tool to annotate Aspergillus compounds using m/z data obtained by MS instrumentation."
--> This statement is incorrect! The NPAtlas (
https://www.npatlas.org/joomla/index.php --> https://pubs.acs.org/doi/10.1021/acscentsci.9b00806) provides 2034 manually curated Aspergillus metabolites with their elemental formulas and accurate masses. This is also double the amount of the database the authors provide here. Thus, this resource needs to be cited in the manuscript and the authors will need to carefully explained how their database is different from what NPAtlas offers (what is the novelty [i.e., the web-tool to search for the metabolites], and why not simply integrate the resources to prevent database fragmentation and retrieve the metabolites from NPAtlas). How many metabolites are reported by the authors and not by NPAtlas? Also, the authors could add the missing Aspergillus metabolites in their database and vice versa? Also, add NPAtlas into the Venn Diagram of Figure 3.

We appreciate this comment. However, the NPAtlas does not provide a m/z search. It is a very useful resource containing complete information about natural products, but the searches are based on the monoisotopic mass; hence a manual transformation of the m/z obtained in mass spectroscopy to monoisotopic mass has to be performed. Furthermore, the searches need to be performed mass by mass since NPAtlas does not provide a batch search. The amount of metabolomic data makes this task a hectic process, therefore an m/z search service that takes into account the potential adducts and permits a batch search, such as our service, is very valuable for the researchers.

We believe that the inclusion of the NP Atlas data into the original Venn diagram does not suit, since this is a comparison of tools providing batch searches that takes into account the adduct formation. Nevertheless, we have included a second Venn diagram with the coverage between the databases knapsack, NP Atlas and A2MDB, the main databases containing metabolites present in genus Aspergillus. We acknowledge again the reviewer for this comment, it points out some aspects that we hope that are clearer in the new version of the paper.

Application of the new database:
--> Whilst this reviewer can see the potential use cases of the database (whether or not integrated with NPAtlas), showing an example of how it is applied to a fungal extract seems lacking from the manuscript. It would also make a lot of things implicitly mentioned in the manuscript more explicit. This would also add a nice Figure where a chromatogram with numerous annotated Aspergillus metabolites could be displayed that would not have had useful candidates in other open databases. The authors could take one dataset of their own work, or get a public dataset run on Aspergillus extracts. If MS/MS data of some or several features is available, those annotations could be verified as well and added as novel database compounds to public spectral libraries - which would constitute as a clear contribution from this manuscript. It would serve as an input for a nice graphical summary as well.

Thank you very much for this comment. We have provided an example of using the database. We have taken a set of features from 26 reference standards previously detected in Aspergillus nidulans and Aspergillus fumigatus and we have identified them using different general metabolomic databases providing m/z searches, NP Atlas after a manual transformation to monoisotopic masses, and the Aspergillus Metabolome database. We do hope that it proves the utility of the service when performing MS metabolomics.

English language
--> Language needs polishing at several instances - especially grammar (use of present and paste tense). Especially so in the abstract as well as throughout the manuscript.

Thank you for pointing this out. We have thoroughly reviewed the English grammar.

Introduction

"Aspergillus fumigatus is the 42 major cause of disease in humans followed by Aspergillus flavus, Aspergillus niger, Aspergillus terreus, Aspergillus nidulans and several members of Aspergillus section Fumigati [13,16,17]."
--> How was this ranking done? And by whom? By the authors? This needs some clarification in the manuscript.

Thank you for this comment. The ranking was not performed by the authors, it is taken from a literature reference (13, https://www.ncbi.nlm.nih.gov/pmc/articles/PMC4315914/) and supported by the references 16 and 17. Nevertheless, we have rewritten the sentence to make it clearer:

"Metabolomics is considered the omic" --> Rephrase: Metabolomics is considered to be the omics discipline

The phrase was rewritten, thank you for the correction.

"However, one of the main challenges in untargeted metabolomics is metabolite identification." 
--> this statement needs some references.

The next references have been included in the paper:

Blazenovic, I; Kind, T.; Ji, J; Fiehn, O. Software Tools and Approaches for Compound Identification of LC-MS/MS Data in Metabolomics. Metabolites 2018, 8, 31.

Nguyen, D.H.; Nguyen, C.H.; Mamitsuka, H. Recent advances and prospects of computational methods for metabolite identification: a review with emphasis on machine learning approaches. Brief. Bioinform. 2018, 20, 2028-2043.

Monge, M.E.; Dodds, J.N.; Baker, E.S.; Edison, A.S.; Fernández, F.M. Challenges in Identifying the Dark Molecules of Life.         Annu. Rev. Anal. Chem. 2019, 12, 177-199.

"Finally, the results of this paper are discussed and conclusions are drawn."
--> instead of this "empty" statement, the authors could actually shortly summarize the main results, findings, and conclusions here....

Thank you for the correction. We have summarized the main results, and conclusions at the end of the introduction. The service created to perform metabolite annotation on Aspergillus studies will increase the Aspergillus metabolome coverage compared to the main metabolomic databases used in untargeted metabolomics such as METLIN or NPAtlas.

Material and Methods

"to add metainformation relative to scientific publications where metabolites (both those found in the bibliographic review and those imported automatically) are discussed. Among the included metainformation from the literature review are also associations between metabolites and the Aspergillus species where they are present."
--> was this metainformation also computer-readable stored? More detailed information at this stage on the actual content, how it could be used, and the purposes, is needed here.

Thank you again for help us clarifying what we have done. We have rewritten the sentence. The metainformation is computer-readable through a RESTful API or the mysqldump file. The actual use of the metainformation has been described. This information is useful for researchers that can check if the putative metabolites associated to the features obtained by MS means have been previously detected in similar organisms (information regarding in which Aspergillus species these compounds have been previously identified) and in which scientific publications these compounds are mentioned. The full information about compounds is provided through a RESTful API and in the MySQL backup database.

"we searched for the name in the 164 CMM database, which contains 186,638 compounds."
--> this needs a ref to the "CMM" database. And also explain the abbreviation "CMM" here!

The abbreviation CMM is defined and cited in line 105.

"(see Supplementary Information, file “SI1_fullERModel.mwb”)"
--> it is unclear what is present in this file, or how it can be opened
.

Thank you noticing that. We wrongly assumed that the format was known. It is a technical detail and now it is stated in the manuscript and the github repository. Information about what contains this file and how to open it has been added the first time that it is mentioned (line 184). This file is a MySQL model that can be open using MySQL Workbench to navigate through all the entities present in CMM.

"The information about the biological activity of the compounds, when available, has also been included as meta-information in the database."
--> How was this done? Did the authors make several categories? Was a particular ontology used here? And is it computer-readable? Can you search based on bioactivity? Could you provide an overview in the manuscript of this? This could be of added value to the community by linking the activity to the structures and then to omics data obtained from those structures.

Thank you for this wise comment. The purpose of this project is not to create a new ontology or taxonomy about biological activities, which is out of the scope of the manuscript. The biological activity corresponds to a free text field to give some information to the user based on the literature review. In the future, we will consider upgrading the project and including/creating a new taxonomy for biological activities of compounds. In the manuscript a clarification has been done, indicating that it is a free text that we have collected for molecules from a literature reference, when available.

“The biological activity consists on free text description based on the literature review performed.”

Figure 3
--> The Venn diagram implies that some databases contain "unique" Aspergillus metabolites (i.e., on the right hand side the circles are outside the main large circle), whereas in practice all metabolites are part of the author's database, right? This should be clarified as it is confusing now.
In addition, the Figure is not that "space-efficient" and the authors could consider a different representation of the same information.

It is confusing indeed. It was a problem with the tool used to represent the Venn diagram. It did not represent the actual data obtained. As the reviewer has noticed, the databases here presented do not contain unique Aspergillus metabolites already known, since they would have been incorporated to the CMM database. The Venn Diagram has been updated addressing this issue and a second figure comparing the coverage of the main databases containing information about Aspergillus Compounds (KNapSaCK, NPAtlas and A2MDB) has been included.

Discussions and conclusions

"Therefore, we cannot include these potential Aspergillus compounds in the Aspergillus Metabolome Database (note that if this calculation was possible, the automatic incorporation of these compounds into the Aspergillus Metabolome Database would also be possible)."
--> What do the authors mean with "calculation" here? This is unclear and needs rephrasing.

Thank you very much for this comment as well. We have rephrased to:

“Due to the lack of metadata regarding which compounds are present in some Aspergillus species in the general metabolomic databases, it is not possible to know if there are compounds identified in Aspergillus species present in them that have not been included in the Aspergillus Metabolome Database. This also means that, if such compounds existed, it is not possible to include them in the Aspergillus Metabolome Database.”